# Frequency, geographical distribution and outcomes of pit viper bites in Malaysia consulted to Remote Envenomation Consultancy Services (RECS) from 2017 to 2020

**Reza Murad Qamruddin[1], Ruth Sabrina Safferi[2], Zainalabidin Mohamed@Ismail[3], Mohd Shukruddeen Salleh[4], Muhammad Nadzmi Hadi Abd Hamid[5], Vera Effa Rezar Frederic Ng[5], Wan Chee Goh[5], Ahmad Khaldun Ismail[5,6]***

**1** Emergency and Trauma Department, Hospital Melaka, Malaysia, **2** Emergency and Trauma Department, Hospital Raja Permaisuri Bainun, Ipoh, Perak, Malaysia, **3** Emergency and Trauma Department, Hospital Tengku Ampuan Afzan, Kuantan, Pahang, Malaysia, **4** Emergency and Trauma Department, Hospital Sultan Ismail Petra, Kuala Krai, Kelantan, Malaysia, **5** Department of Emergency Medicine, Faculty of Medicine, Universiti Kebangsaan Malaysia, Kuala Lumpur, Malaysia, **6** Hospital Canselor Tuanku Muhriz, Jalan Yaacob Latif, Bandar Tun Razak, Kuala Lumpur, Malaysia

* khaldun_ismail@yahoo.com

## Abstract

Not all pit viper species are present in every state of Malaysia and their distribution varies according to altitude. There is limited information on pit viper bite incidence and its geographical distribution. This was a cross-sectional study of confirmed pit viper bite cases referred to Remote Envenomation Consultancy Services (RECS) from January 2017 to December 2020. Data was collected following the approval of institutional research ethics committee. Universal sampling methods were used. Confirmed pit viper bite cases in each state, geographical location and the antivenom used were reported. A total of 523 confirmed pit viper bite injuries occurred over the 4-year study period. The majority were Malaysians, male and young adults. Most were non-occupational related (83.9%) and involved the upper limbs (46.8%). The commonest pit viper species involved was *Trimeresurus purpureomaculatus* (23.7%). Green pit viper antivenom (GPAV) was the most frequent antivenom used (*n* = 51) with the majority of patients requiring only one dose (3 vials). This study provides a better appreciation of indigenous pit viper species distribution for each state and reflects the requirement of appropriate antivenom to be stocked in each state or district hospital.

## Author summary

There is limited information on pit viper bite in Malaysia which may also reflects poor awareness and documentation. This study analysed RECS consultation log, frequency, species involved, geographical distribution and clinical management outcomes of pit viper bites in Malaysia from 2017–2020. RECS provides reliable and rapid information on

**Data Availability Statement:** All relevant data are within the manuscript and its Supporting Information files.

**Funding:** The author(s) received no specific funding for this work.

**Competing interests:** The authors have declared that no competing interests exist.

snake species identification, location of incidence and the appropriate choice of anti-venom when indicated. Pit vipers in Malaysia are classified into several genus and their geographical distribution is not homogenous. Not all pit vipers cause significant envenomation to human and requiring antivenom. All three types of antivenoms appropriate for pit viper envenomation were manufactured and imported from Thailand. From this study, the optimal management and appropriate antivenom stock for each hospital can be determined.

## Introduction

Snake related injury (SRI), especially snakebite envenomation, is a medical emergency. The actual burden from snakebite envenomation in affected South-East Asian countries is poorly understood mainly due to lack of properly designed population surveys looking at incidence, morbidity, mortality and access to antivenom [1–4]. Literature searches on snakebite epidemiology in Malaysia are scarce and most diagnoses were unverified, especially regarding the identification of the snake species. Most studies are limited to individual hospitals in different states in Malaysia [5–11]. The diagnoses were recorded at admission registration or discharge records and are broadly categorised as 'cobra' or 'viper' bite. Almost all previous studies of snakebite in Malaysia do not include the cases from East Malaysia (the states of Sabah and Sarawak).

Part of the challenge is because SRI is not classified as a notifiable disease in Malaysia. It has been estimated that snakebite incidence in Malaysia is 400–650 per 100,000 population per year with an annual mortality of 0.2 per 100,000 population [12]. The majority of snake bite cases are from non-venomous snakes but venomous snake bites do cause significant morbidity and mortality if treatment, especially antivenom (AV) therapy, is delayed. However, the true burden of human suffering from snake envenomation remains uncertain in Malaysia. It is therefore critical to improve data acquisition and analysis for SRI.

In Malaysia, pit vipers can be categorised according to their habitat—terrestrial, arboreal or both. Currently there are 16 pit viper species in Malaysia. The geographical distribution of each pit viper species is not homogenous throughout the country and may vary according to altitude [13–22]. Documented pit viper species include *Trimeresurus purpureomaculatus, T. hageni, T. nebularis, T. malcomi, T. sabahi fucatus, T. borneensis, T. buniana, T. sabahi sabahi, T. sumatranus, T. venustus, T. wiroti, Tropidolaemus subannulatus, Tropidolaemus wagleri, Calloselasma rhodostoma, Garthius chaseni,* and *Ovophis convictus* [13,14,16]

Malaysia Biodiversity Information System (MyBIS) is a one-stop repository of biodiversity data in Malaysia that also provides the information platform on biological diversity. In 2017, MyBIS and Remote Envenomation Consultancy Services (RECS) collaborated to record all RECS consultation data into a systematic registry called MyBIS Toxinology module [23]. The process involved filing of individual cases into the digital recording system. Various information such as snake species and incident location were tabulated and analysed. RECS was developed to assist healthcare professionals providing clinical management for bites/stings envenoming from venomous animals and poisoning from naturally occurring toxins [23,24]. The main objective of RECS is to improve outcomes by optimizing and advocating appropriate treatment modalities at every level of clinical management.

The objective of this study is to determine the frequency of pit viper bites consulted to RECS, the geographical distribution and the appropriate antivenom used for each case.

## Methods

This is a retrospective cohort study of confirmed pit viper cases in Malaysia consulted to RECS from January 2017 to December 2020. Data was collected following the approval of institutional research ethics committee Faculty of Medicine, National University of Malaysia (UKMFPR.19/244) and RECS. All data collected from RECS database were kept anonymous and confidential. A universal sampling method was used whereby all case details were collected from RECS consultation log and case record. This study did not involve hypothetical testing thus no sample size calculation was required. The data were analysed using SPSS 25. Only confirmed pit viper species bite cases were included in this study. The species confirmation was verified by RECS experts during each consultation based on the specimen brought to the hospital or the picture of the actual specimen taken. Verified pit viper bite cases in each state, geographical location, and the antivenom used were reported.

## Results

There was a total of 4005 consultations with 84.2% ($n$ = 3375) SRI. Unidentified SRI was 56.7% ($n$ = 1912). Of 1463 identified cases, 35.7% ($n$ = 523) were confirmed pit viper bite injuries. The majority of patients were Malaysians, male and young adults between 19 and 59 years old (Table 1). The median age was 38 years old (IQR = 23,54). Most incidences were non-occupational related ($n$ = 439, 83.9%). The most frequent body part affected was the upper limb primarily the hand ($n$ = 246, 46.8%) followed by the foot ($n$ = 141, 26.8%). Two cases had multiple bites and involved more than one body area.

The most common identified pit viper species involved was *Trimeresurus purpureomaculatus* ($n$ = 124, 23.7%) followed by *Tropidolaemus subannulatus* ($n$ = 121, 23.1%), *Tropidolaemus wagleri* ($n$ = 84, 16.1%) and *Calloselasma rhodostoma* ($n$ = 58, 11.1%) (Table 2). There were no confirmed bite incidences recorded from *T. buniana* and *T. malcomi* throughout the study period. There was one exotic pit viper bite case, *Trimeresurus flavomaculatus*, which had been illegally imported from the Philippines. Most incidents occurred during daytime with a slight increase in incidents between October and December (Fig 1). Most cases were referred from the state of Sarawak ($n$ = 124, 23.7%), followed by Perak ($n$ = 115, 22.0%) and Pahang ($n$ = 78, 14.9%) (Fig 2). Not all pit viper species are present in every state in Malaysia (Fig 3). The distribution of pit viper species according to States and Gazetteer for the year 2017–2020 is provided in the supplementary table (S1 Table).

The confirmed pit viper species causing significant envenomation and received antivenom were *C. rhodostoma*, *O. convictus*, *T. purpureomaculatus*, *T. borneensis*, *T. hageni*, *T. nebularis*, *T. sabahi sabahi*, *T. sabahi fucatus*, *T. sumatranus*, and *T. wiroti*. Green pit viper antivenom (GPAV) from Queen Saovabha Memorial Institute (QSMI), Thailand was the most frequent antivenom used (Fig 4). Most cases required one dose (3 vials) of antivenom. Of the total of 1912 cases of unidentified SRI over the study period, 7.4% ($n$ = 142) received GPAV, *Calloselasma rhodostoma* antivenom CRAV or Hemato Polyvalent antivenom (HPAV) from QSMI. Two cases were given *Naja Kaouthia* antivenom (NKAV) from QSMI prior to RECS consultation, and subsequently received the appropriate pit viper AV following consultation. Of the 523 confirmed cases only one (0.2%) patient required fasciotomy and amputation of left middle finger. No death was documented.

## Discussion

In our knowledge, this is the first study in Malaysia that highlighted the incidence, geographical distribution and outcome of bites from confirmed pit viper species consulted to RECS. Similar, documentation of envenomation due to bites of snakes identified by experts is needed

**Table 1. Demographic data of confirmed pit viper bite patients consulted to RECS from 2017–2020.**

| Categories | Frequency (n) | Percentage (%) |
|---|---|---|
| **Age** | | |
| 0–12 | 55 | 10.5 |
| 13–18 | 40 | 7.6 |
| 19–59 | 341 | 65.2 |
| >60 | 86 | 16.4 |
| Not documented | 1 | 0.2 |
| **Gender** | | |
| Male | 399 | 76.3 |
| Female | 123 | 23.5 |
| Not documented | 1 | 0.2 |
| **Nationality** | | |
| Malaysian | 444 | 84.9 |
| Non-Malaysian | 78 | 14.9 |
| Not documented | 1 | 0.2 |
| **Activity** | | |
| Occupational | 80 | 15.3 |
| Non occupational related | 439 | 83.9 |
| Not documented | 4 | 0.8 |
| **Part of body affected** | | |
| Not documented | 1 | 0.2 |
| Head and neck | 7 | 1.3 |
| Arm | 12 | 2.3 |
| Forearm | 18 | 3.3 |
| Hand | 246 | 46.8 |
| Thigh | 11 | 2.1 |
| Leg | 37 | 6.9 |
| Ankle | 49 | 9.4 |
| Foot | 141 | 27.0 |
| Buttock | 3 | 0.6 |

in this country and elsewhere affected by snakebites. The robust statistical data on snakebite envenomation is lacking and cases reported to health ministry by clinics and hospitals are an underestimate of the actual burden because many victims never reach primary care facilities [25,26]. Further, the Malaysian literature on proven snakebite epidemiology is scarce [12,17]. These are challenges to assess the true impact of snakebite envenomation in Malaysia. Our report of incidence and geographical distribution of expert-identified pit viper species can be implied for the proper snakebite management in Malaysia.

The checklist of pit viper species (Table 2) corresponds to the recent review of the terrestrial venomous snakes of Malaysia including those from the states of Sabah and Sarawak [15]. Overall, in addition to the viperid snakes, colubrid snake species in Malaysia, for example, the Keelbacks, *Rhabdophis* species, can cause significant coagulopathy [12,17].

Sixteen species of pit vipers have been described in Malaysia with geographical distribution data based on field sampling [13,14,16,18]. Most pit viper species in Malaysia are ambush predators with good camouflage. Most of the pit vipers bite cases in this study involved local citizens and occurred during non-occupational activity possibly because pit vipers are mostly nocturnal and outdoor activities are mainly performed during daytime. It is uncommon to

**Table 2. The identified pit viper species involved in the bite incidents consulted to RECS from 2017–2020.**

| Pit viper species | Frequency (*n*) | Percentage (%) |
|---|---|---|
| *Calloselasma rhodostoma* | 58 | 11.1 |
| *Garthius chaseni* | 1 | 0.2 |
| *Ovophis convictus* | 17 | 3.3 |
| *Trimeresurus borneensis* | 28 | 5.4 |
| *Trimeresurus flavomaculatus** | 1 | 0.2 |
| *Trimeresurus hageni* | 24 | 4.6 |
| *Trimeresurus nebularis* | 13 | 2.5 |
| *Trimeresurus purpureomaculatus* | 124 | 23.7 |
| *Trimeresurus sabahi fucatus* | 26 | 5.0 |
| *Trimeresurus sabahi sabahi* | 4 | 0.8 |
| *Trimeresurus sumatranus* | 15 | 2.9 |
| *Trimeresurus venustus* | 1 | 0.2 |
| *Trimeresurus wiroti* | 6 | 1.1 |
| *Tropidolaemus subannulatus* | 121 | 23.1 |
| *Tropidolaemus wagleri* | 84 | 16.1 |

*Note: *Trimeresurus flavomaculatus* reported in this case study is non-indigenous and illegally imported from the Philippines

find pit vipers entering houses, and in this study only 2.9% of cases occurred indoors. Most of the pit vipers in Malaysia are arboreal and belonging to the *Trimeresurus* complex species. There were no confirmed cases recorded from two species, *T. buniana* and *T. malcomi*, during the study period, but that does not mean that these two species are not present in Malaysia. *T. buniana* is found only on Tioman island and *T. malcomi* are found on the highlands of Sabah and Sarawak. Both locations have low density of human population and the habitat is still relatively well preserved.

This study showed that the upper limbs were more frequently involved than the lower limbs. This was probably because most of the patients were actively using their hands during

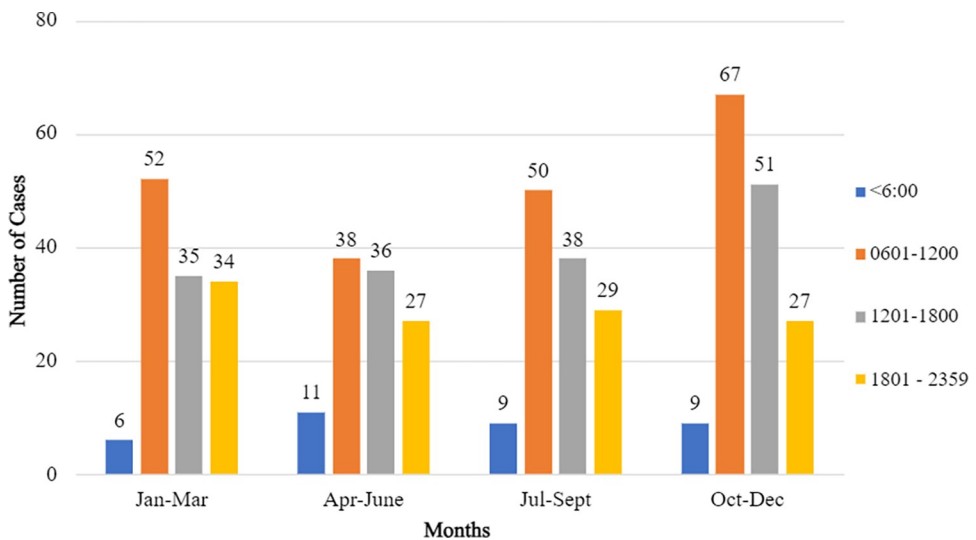

**Fig 1. Frequency of cases for each month and the time of incident at quarterly intervals of the day.**

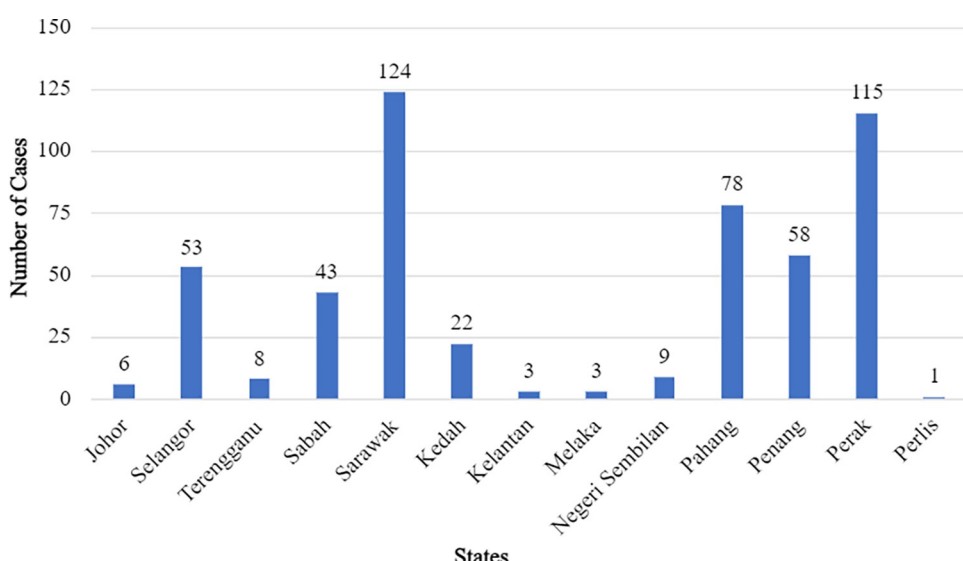

**Fig 2. Frequency of pit viper bite injuries consulted to RECS from each state in Malaysia from 2017 to 2020.**

non-occupational related activities close to the ground or above ground. This is in contrast to earlier studies on snake bite in Malaysia that identified the lower limbs to be more frequently affected [5–11]. This contradiction may be influenced by different snake species involved. The involvement of upper limbs in this study may have been influenced by the activities prior to the incident and the location of the snakes. The common species identified in the other studies were *Naja kaouthia*, *Naja sumatrana*, and *Calloselasma rhodostoma*. These species inhabit

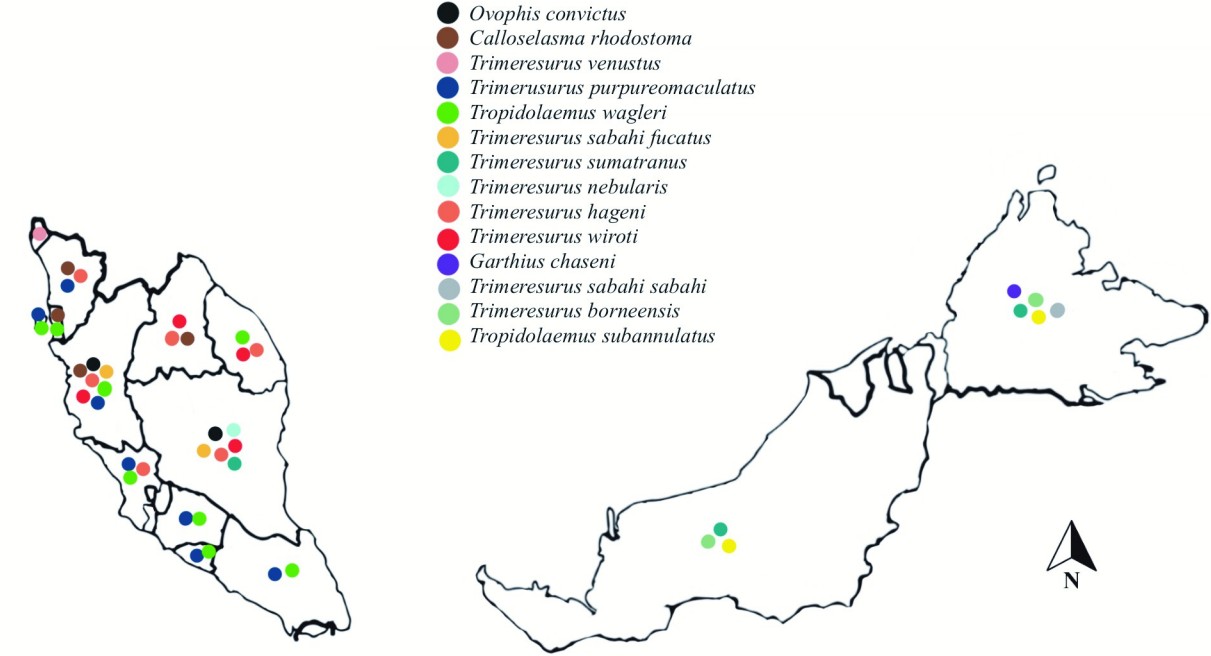

**Fig 3. Geographical distribution of pit viper species consulted to RECS in each state in Peninsular Malaysia and East Malaysia from 2017–2020.** Base map and data from OpenStreetMap and OpenStreetMap Foundation. (OpenStreetMap contributors, https://www.openstreetmap.org/#map=6/4.226/108.237).

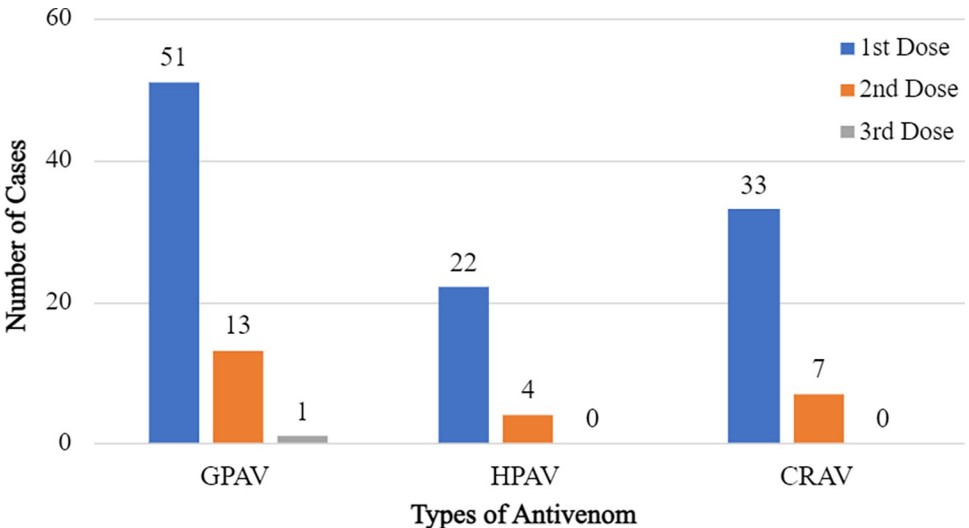

**Fig 4. Types and frequency of pit viper antivenom administered from 2017–2020.**

lowland forest with an elevation of less than 820m above sea level and frequently found in open environment such as agricultural fields and plantations. They are mainly terrestrial in nature and the cobras are fast moving snakes that can rear up one-third of its body length. The majority of bites from these species were found to be accidental in nature either by stepping on the snake, getting too close to it or did not move away fast enough.

Snake related injury is a neglected public health issue in many tropical and subtropical countries [25,26]. Health systems, socio-economic and cultural factors directly influence the outcome of snakebite injury. A study in Thailand with a similar health system and socio-economic background, reported a decline in incidence and low case fatality of 0.05% with majority of cases involving pit vipers [1–4]. Despite having similar medically important snake species Malaysia does not manufacture its own antivenom and does not stock antivenom for non-indigenous snake species. All of the antivenom for indigenous species in Malaysia is imported from Thailand and Australia [12,17].

Antivenom is certainly required for pit viper bite envenomation when indicated. However, not all types of pit viper antivenom appropriate for Malaysia is required to be stocked in every hospital. This study has shown that there is specific geographical distribution of pit viper species between the states in Peninsular Malaysia, Sabah and Sarawak. Each hospital can therefore decide on the appropriate antivenom to optimise the individual hospital stock based on the prevalence of each pit viper envenomation treated [12,16,17,27,28].

Green pit viper AV is the most utilized for treating envenomation from *Trimeresurus* complex species and its appropriateness to be stocked in Malaysian hospitals was reviewed and included in the Ministry of Health snakebite management guidelines [17, 27]. The geographical distribution shows that the Malayan pit viper mainly occurs in the northern states of Peninsula Malaysia, thus CRAV is only appropriate to be stocked in the hospitals within these states. However, a small number of confirmed Malayan pit viper bite cases were documented within a small area in the south of capital city Kuala Lumpur. Therefore, a few hospitals within this area may need to restock CRAV. Based on a recent study, no antivenom is required for bites from *Tropidolaemus* species in the Peninsular and East Malaysia [28].

Even though the majority of identified pit viper bite patients did not receive AV, some of the unidentified pit viper bite patients did receive AV based on the syndromic approach and

close clinical observation. This was usually due to derangement in the coagulation profile, thrombocytopenia, rate of proximal progression of the oedema and pain score progression. Ultrasonography of the affected area or limb can help to locate and verify the affected site, depth of involvement as well as the speed of progression of the oedema [29–33]. Recent studies in Taiwan shows the promising role of point of care ultrasonography (POCUS) in minimising unnecessary surgical intervention and optimizing appropriate AV usage [32,33].

The outcome of pit viper bite envenomation in this study appears to be favourable with extremely low frequency of morbidity or mortality. This could be partly due to the standard of care provided and by the current state of health system with remote clinical consultation with experts in the field. However, Malaysia faces an increasing threat from illegal importation of potentially exotic animals including venomous snakes from ASEAN region and other continents. There were cases of snakebite envenomation from *Trimeresurus flavomaculatus*, *Crotalus atrox*, *Crotalus horidus* and *Bitis arietans*. They have been referred to RECS over the past 10 years. All were from the Viperidae family. Some of these cases have resulted in morbidity and death [34].

## Study limitations

The true prevalence of pit viper species envenomation in Malaysia is likely to be an underestimate as only cases referred to RECS were included in this study. Nevertheless, this study provides a good representative sampling of pit viper bites as all diagnoses were verified by experts in the field and patient's details were well documented with serial pictures and progress notes.

## Conclusion

Pit viper bite is common in Malaysia. A distinct variation in distribution patterns of pit viper species exists. This distinction and the potential of envenomation requiring antivenom for selected pit viper species suggests a judicious stocking and using antivenom in hospitals in different states of Malaysia. There is importance of similar and appropriately designed study to record and document proven snakebites due to other snake family in Malaysia. This study finding also promotes appropriate health seeking awareness and optimal patients' care consulting with the relevant experts. This can be further enhanced by including SRI as a notifiable disease in Malaysia.

## Supporting information

**S1 Table. The distribution of pit viper species according to States and Gazetteer for the year 2017–2020.**
(DOCX)

## Acknowledgments

We would like to thank Dr Azhana Hassan, Dr Anisah Adnan, Dr Razak Daud, Dr Mohd Zaki Fadzil Senek, Dr Noredelina Mohd Noor, Dr Yvonne Teo Chiang Hoon, Dr Anas Amri Hashim and Dr Aida Nur Sharini Mohd Shah for their invaluable support as Malaysian Remote Envenomation Consultation Services consultants. We thank Dr Chan Xin Yi, Miss Nurfarhana Hizan and Miss Nur Hazwanie Abd Halim, Miss Nur Alissa Ariff, Miss Nur Syafiqah Abdul Samat and Professor Colin Robertson for technical support.

## Author Contributions

**Conceptualization:** Ahmad Khaldun Ismail.

**Data curation:** Reza Murad Qamruddin, Ahmad Khaldun Ismail.

**Formal analysis:** Reza Murad Qamruddin, Ahmad Khaldun Ismail.

**Funding acquisition:** Ahmad Khaldun Ismail.

**Investigation:** Reza Murad Qamruddin, Ahmad Khaldun Ismail.

**Methodology:** Reza Murad Qamruddin, Ahmad Khaldun Ismail.

**Project administration:** Ahmad Khaldun Ismail.

**Resources:** Ahmad Khaldun Ismail.

**Supervision:** Ahmad Khaldun Ismail.

**Validation:** Ahmad Khaldun Ismail.

**Writing – original draft:** Reza Murad Qamruddin, Ahmad Khaldun Ismail.

**Writing – review & editing:** Reza Murad Qamruddin, Ruth Sabrina Safferi, Zainalabidin Mohamed@Ismail, Mohd Shukruddeen Salleh, Muhammad Nadzmi Hadi Abd Hamid, Vera Effa Rezar Frederic Ng, Wan Chee Goh, Ahmad Khaldun Ismail.

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
