## [Decision Letter · Decision Letter 0]

11 Jun 2023

Dear Dr Ismail,

Thank you very much for submitting your manuscript "Frequency, Geographical Distribution and Outcomes of Pit Viper Bite Envenomation in Malaysia" for consideration at PLOS Neglected Tropical Diseases. As with all papers reviewed by the journal, your manuscript was reviewed by members of the editorial board and by several independent reviewers. In light of the reviews (below this email), we would like to invite the resubmission of a significantly-revised version that takes into account the reviewers' comments. 

We cannot make any decision about publication until we have seen the revised manuscript and your response to the reviewers' comments. Your revised manuscript is also likely to be sent to reviewers for further evaluation.

Sincerely,

Wuelton M. Monteiro, Ph.D.

Section Editor

Wuelton Monteiro

Section Editor

Reviewer's Responses to Questions

**Key Review Criteria Required for Acceptance?**

**Methods**

-Are the objectives of the study clearly articulated with a clear testable hypothesis stated?

-Is the study design appropriate to address the stated objectives?

-Is the population clearly described and appropriate for the hypothesis being tested?

-Is the sample size sufficient to ensure adequate power to address the hypothesis being tested?

-Were correct statistical analysis used to support conclusions?

-Are there concerns about ethical or regulatory requirements being met?

Reviewer #1: -No

-yes

-Not relevant

-Not relevant

-Not given

-No

Reviewer #2: It is ok to me. Please, see the attached file.

Reviewer #3: Methods are straight forward. To provide ethics approval regulatory body and approval code.

**Results**

-Does the analysis presented match the analysis plan?

-Are the results clearly and completely presented?

-Are the figures (Tables, Images) of sufficient quality for clarity?

Reviewer #1: -Analysis plan not mentioned

-Yes

-Figure labels (letters) are not clearly visible

Reviewer #2: yes.

Reviewer #3: Results are analyzed according to plan.

Results are clearly presented but can be furnished with findings for outcome.

**Conclusions**

-Are the conclusions supported by the data presented?

-Are the limitations of analysis clearly described?

-Do the authors discuss how these data can be helpful to advance our understanding of the topic under study?

-Is public health relevance addressed?

Reviewer #1: -yes

-yes

-No

-No

Reviewer #2: Yes.

Reviewer #3: Conclusions are made in line with the findings. The proposed use of one of the antivenom for GPAV (which is a mono-specific antivenom) for the diverse pit viper species should be supported by previous works that show cross-reactivity and cross-neutralization at least for some of the species implicated in the cases reported.

**Editorial and Data Presentation Modifications?**

Reviewer #1: (No Response)

Reviewer #2: Minor revision is needed.

Reviewer #3: -

**Summary and General Comments**

Reviewer #1: This study provides important conclusions to the health care system in order to plan proper management of snakebites, particularly having sufficient stocks of antivenom. 

But, there are several issues in the manuscript.

 Method is not clear. Authors have not mentioned sufficient information in this section. Therefore, it should be rewritten including the following concerns,

-although it was mentioned in Introduction about a collaboration of MyBIS with RECS, it was not mentioned in Method how data were collected from the system for this retrospective study. Authors should mention clearly the details regarding this information system.

-Mention about the administrative approval 

-Usually, in hospital clinic records, species names have not been recorded and common names are mentioned instead e.g. Green pit viper, Russell’s viper etc. unless if a well-planned prospective study is done. Therefore, in this situation, how did authors identify species of pit vipers

Heading

It is better to remove the word envenomation and the suggested heading is,

Frequency, Geographical Distribution and Outcomes of Pit Viper Bites in Malaysia

Introduction

Line 54-56

write short sentences

Line 56-57

meaning is not clear

Line 68-72

How many pit viper species in Malaysia? Are there endemic species?

Methods

Line 88

Mention ERC reference No.

Line 89

Considering the retrospective nature of this study, explain universal sampling method of the study

Line 92-93

As this is a retrospective study, explain “The species confirmation was verified by RECS experts during each consultation based on the specimen brought to the hospital”

Are all these data available in the patients’ clinical records?

Line 93

How species identification was possible by using a picture?

Results

Table 2

How the species level identification was done in this study?

Line 132-133

What are OSMI and NKAV?

Discussion

184-185

….no antivenom is required for bites from Tropidolaemus

species in the peninsula and east Malaysia.

Line 192

US??

Line 296

…diagnoses….

References

Please replace very old references e.g 4, 7, 8 with recent ones

Tables

Table 1

It is not clear following contents

Not documented 1

median 38

Minimum 0

Max 85

Please use a clear table format

Figures

Letters are not clear (blurred) in all figures

Reviewer #2: see the attached file

Reviewer #3: Authors reported a study entitled: Frequency, Geographical Distribution and Outcomes of Pit Viper Bite Envenomation in Malaysia. This is an important work as data on snakebite envenomation is lacking in the country. However, there are several issues with the write-up which authors should try to address for improvement.

Comment: From the public health perspective, this is not a typical epidemiological study and is solely based on consultancy from doctors/hospitals to a group of network (RECS), there is a confusion over how representative of the overall data is for the whole country (Malaysia), as suggested by the title. It is good that authors acknowledged this a limitation (see section on limitation). However, the following suggestions should be addressed: 

(1) For clarity, the title should be modified accordingly to reflect the nature of the work. May include "analysis based on 523 pit viper envenomation cases consulted to RECS (2017-2020)" as part of it. 

(2) The "outcomes" is not well defined. Was this referred to the clinical outcome as in both syndrome/toxicity and treatment? Readers from afar would be expecting to read these, naturally as the outcomes of those cases following consultancy provided by the RECS team. The article in fact did not report in depth the clinical manifestations of the bites by the respective species, and no much information was provided on the treatment aspect (for instance, how many vials of ?GPAV in average, for each species). These should be considered as part of important data to substantiate this article. 

Introduction is adequate. 

Methodology is straight forward. 

Line 89-90: Data was collected following the approval of

institutional research ethics committee and RECS. -- Please provide name of institution/IRB and ethics code. 

Results, discussion and conclusion:

Main comment and suggestion: 

Line 68-72 and geographical map of pit viper cases consulted to RECS show there are very diverse species of pit vipers responsible for SRI and envenomation in Malaysia, and proposed the use of mainly two antivenoms imported from Thailand for use against these indigenous species in Malaysia. The use of CRAV is clearly for the Malayan pit viper as authors rightly explained. However, GPAV is proposed by authors as the antivenom to be used for other species? See also: Line 129-131, Line 167-170. Although many of these species are given the genus Trimeresurus, Malhotra et al. has shown the phylogenetic divergence in them, for example, there are genus or subgenus like Popeia and Craspedocephalus and Parias etc. I think there are also a number of studies showed the variation in their venom content and toxicity. Authors should include the outcome findings of the treatment (they did mention for T. purpureomaculatus, treated with GPAV 3 vials) for the various species consulted to them. It is understood that clinical trial is almost impossible but there have been several experimental or preclinical studies in animals that demonstrated the cross-reactivity or cross-neutralisation activity of GPAV for the various Trimeresurus pit vipers, in particular T. purpureomaculatus since this is the most common biting pit viper species in their consultancy record. These studies should be referenced in making the call for the clinical use of GPAV for the non-species specific bites.

PLOS authors have the option to publish the peer review history of their article (what does this mean?). If published, this will include your full peer review and any attached files.

Reviewer #1: No

Reviewer #2: Yes: Deb Prasad PANDEY

Reviewer #3: No
---

## [Decision Letter · Decision Letter 1]

25 Jul 2023

Dear Dr Ismail,

Thank you very much for submitting your manuscript "Frequency, Geographical Distribution and Outcomes of Pit Viper Bites in Malaysia Consulted to Remote Envenomation Consultancy Services (RECS) from 2017 to 2020" for consideration at PLOS Neglected Tropical Diseases. As with all papers reviewed by the journal, your manuscript was reviewed by members of the editorial board and by several independent reviewers. In light of the reviews (below this email), we would like to invite the resubmission of a significantly-revised version that takes into account the reviewers' comments. 

We cannot make any decision about publication until we have seen the revised manuscript and your response to the reviewers' comments. Your revised manuscript is also likely to be sent to reviewers for further evaluation.

Sincerely,

Wuelton M. Monteiro, Ph.D.

Section Editor

Wuelton Monteiro

Section Editor

Reviewer's Responses to Questions

**Key Review Criteria Required for Acceptance?**

**Methods**

-Are the objectives of the study clearly articulated with a clear testable hypothesis stated?

-Is the study design appropriate to address the stated objectives?

-Is the population clearly described and appropriate for the hypothesis being tested?

-Is the sample size sufficient to ensure adequate power to address the hypothesis being tested?

-Were correct statistical analysis used to support conclusions?

-Are there concerns about ethical or regulatory requirements being met?

Reviewer #1: -Not applicable

-Yes

-Not applicable

-Not applicable

-Yes

-No

Reviewer #2: -Are the objectives of the study clearly articulated with a clear testable hypothesis stated?

# yes

-Is the study design appropriate to address the stated objectives?

# yes

-Is the population clearly described and appropriate for the hypothesis being tested?

# This is a descriptive study. They did not design it for the hypothesis testing.

-Is the sample size sufficient to ensure adequate power to address the hypothesis being tested?

#NA

-Were correct statistical analysis used to support conclusions?

# yes

-Are there concerns about ethical or regulatory requirements being met?

# yes

Reviewer #3: -

**Results**

-Does the analysis presented match the analysis plan?

-Are the results clearly and completely presented?

-Are the figures (Tables, Images) of sufficient quality for clarity?

Reviewer #1: -yes

-yes

-yes

Reviewer #2: -Does the analysis presented match the analysis plan?

# yes

-Are the results clearly and completely presented?

# yes

-Are the figures (Tables, Images) of sufficient quality for clarity?

# yes

Reviewer #3: Refer to comments.

**Conclusions**

-Are the conclusions supported by the data presented?

-Are the limitations of analysis clearly described?

-Do the authors discuss how these data can be helpful to advance our understanding of the topic under study?

-Is public health relevance addressed?

Reviewer #1: -yes

-yes

-yes

-yes

Reviewer #2: This needs revision. I have suggested in the attached file.

Reviewer #3: Refer to general comments.

**Editorial and Data Presentation Modifications?**

Reviewer #1: (No Response)

Reviewer #2: Minor revision is needed before its acceptance.

Reviewer #3: (No Response)

**Summary and General Comments**

Reviewer #1: Most of the previous queries have been addressed. But, there are minor corrections as follows,

Line 52

ASEAN ?

Line 64

snake bite

snakebite (one word)

Line 71-74

It is better to include common name of these species in brackets or

Common name then scientific name in brackets

Also, mention the endemic species 

A recent paper describes on taxonomic updates of Green pit vipers. Please see whether these names (genus) are changed or not

Results

Only % has been mentioned. But, number should be there with % as 

n (%).

Line 102

56.7%3?

Line 104

Median value should follow IQR

Line 76-77

scientific names should be italic

Reviewer #2: Please, see the attached file.

Reviewer #3: In paragraphs line 196-202, and according to authors response to one reviewer's comment that says "The proposed use of one of the antivenom for GPAV (which is a mono-specific antivenom) for the diverse pit viper species should be

supported by previous works that show cross-reactivity and cross-neutralization at least for some

of the species implicated in the cases reported." Authors responded that the necessary references were provided without specifying them. To another comment by the reviewer, authors responded "cross neutralization has been published and are quoted in the Ministry of Health guidelines and several other references provided." However, we could not identify these references in the revised version. The previous studies are obvious in the literature, a large body of studies on cross-neutralization of GPAV (Trimeresurus albolabris mono antivenom) with efficacy testing for various Trimeresurus pitviper species, including wiroti, nebularis, purpureomaculatus, sumatranus etc (these species were documented in table 2) have been reported for Malaysian venoms. Perhaps the authors have overlooked the references in their revision, but they would be crucial to support the outcome reported and the proposed use of monovalent GPAV for other pitviper species (except malayan pitvper) in Malaysia.

PLOS authors have the option to publish the peer review history of their article (what does this mean?). If published, this will include your full peer review and any attached files.

Reviewer #1: No

Reviewer #2: Yes: Deb Prasad Pandey

Reviewer #3: No
---

## [Editor Report · Decision Letter 2]

2 Aug 2023

Dear Dr Ismail,

We are pleased to inform you that your manuscript 'Frequency, Geographical Distribution and Outcomes of Pit Viper Bites in Malaysia Consulted to Remote Envenomation Consultancy Services (RECS) from 2017 to 2020' has been provisionally accepted for publication in PLOS Neglected Tropical Diseases.

Best regards,

Wuelton M. Monteiro, Ph.D.

Section Editor

Wuelton Monteiro

Section Editor

---

## [Editor Report · Acceptance letter]

14 Aug 2023

Dear Dr Ismail,

We are delighted to inform you that your manuscript, "Frequency, Geographical Distribution and Outcomes of Pit Viper Bites in Malaysia Consulted to Remote Envenomation Consultancy Services (RECS) from 2017 to 2020," has been formally accepted for publication in PLOS Neglected Tropical Diseases.

Best regards,

Shaden Kamhawi

co-Editor-in-Chief

Paul Brindley

co-Editor-in-Chief
